

# Aspect Controls on the Spatial Re-Distribution of Snow Water Equivalence in a Subalpine Catchment

Kori L. Mooney[1,2], Ryan W. Webb[1]

[1]Department of Civil and Architectural Engineering and Construction Management, University of Wyoming, Laramie, WY, 82072, USA
[2]Natural Resources Conservation Service, Salt Lake City, UT, 84138, USA

*Correspondence to*: Ryan W. Webb (Ryan.Webb@uwyo.edu)

**Abstract.** Quantifying subalpine snowpack parameters as they vary through time with respect to aspect and position on slope are important for estimating the seasonal storage of snow water resources. Snow depth and density are dynamic parameters that change throughout the progression of the accumulation and melt periods, with direct implications on the distribution of Snow Water Equivalence (SWE) across a landscape. Additionally, changes in density can infer physical processes occurring within the snowpack such as compaction, liquid water pooling, and lateral flow. This study measures snow depth and density throughout a 0.25 km² watershed in northern Colorado USA using L-Band (1.0 GHz) Ground Penetrating Radar (GPR) and coincident depth probing. GPR snow densities were calibrated using bulk densities from snow pits and a SNOTEL station. A physical snowpack model, SNOWPACK, with input from local Remote Automated Weather Station and SNOTEL station produced models of snow depth, snow density, and liquid water content (LWC). The model simulations indicate mid-winter melt events produced LWC on the south aspect that are less present in the north aspect and flat areas. These midwinter melt events resulted in the lateral flow of LWC downslope, and the redistribution of SWE as observed in GPR surveys. Further observations show a steady increase of soil moisture throughout the winter in the flat terrain and ice layer formation on the south aspect snow pits during mid-winter surveys. Other key observations include pooling of liquid water at the base of the north aspect during the later spring season melt phase evidenced by pit observations and GPR transects. We further develop a conceptual model for the aspect controls on the distribution and movement of SWE during the winter and spring seasons. In summary, mid-winter melt events are observed on south aspects, causing a redistribution of SWE downslope while spring melt brings liquid water pooling at the base of north aspects. These differences in snowmelt dynamics based primarily on aspect, providing important processes to consider for spatially and temporally extensive SWE measurements moving forward.

## 1 Introduction

Accurately quantifying snow water equivalence (SWE) can provide valuable insight into storage and flux of water resources. SWE can inform spring and summer streamflow generation (Li et al., 2017), soil moisture levels (Mcnamara et al., 2005), and groundwater recharge (Brooks et al., 2021). Additionally, being able to anticipate the timing and quantity of these fluxes can help predict flooding, drought, streamflow volumes, and reservoir storage (Zeinivand and De Smedt, 2010; Modi et al., 2022;



Bishay et al., 2023). Regional distributions in SWE also impact ecosystem services through surface albedo, effectively cooling earth surfaces and regulating climate (Sturm et al., 2017). All of these contribute to functioning societies and ecosystems, making the accurate, precise, and timely measurement of SWE an essential annual metric (Mankin et al., 2015). Rapidly shifting global patterns in moisture delivery contribute to measuring SWE in snow-dominated catchments even more important

with snowpack parameters changing rapidly in response to shifting weather and climate patterns, even in high elevation snowpacks (Clow, 2010; Nolin et al., 2021). The rapidly changing metrics include snow extent, SWE volume, melt out date, precipitation phase, quantity and magnitude of snowfall events, and surface albedo (Clark et al., 2011; Clow, 2010; Erickson et al., 2005; Painter et al., 2016; Skiles et al., 2018).

SWE has been measured since the early 1900s through manual snow courses, and later using weather station networks like

SNOTEL in the western United States. These sites use snow pillows, snow depth sensors, soil moisture, and precipitation gauges to measure seasonal snow fluxes. With sites scattered across high accumulation areas in the United States, these data provide a statistical estimate of water resources based on the relationship between SWE and streamflow. Forecasts have historically been made based on where the water year fits into the period of record, which gives limited context in accounting for long term trends in hydroclimate and deviation from climate stationarity (Sturm et al., 2017; Bales et al., 2006). Thus, the

expansion of snowpack monitoring is necessary to account for spatial and temporal variability found in mountainous environments (Painter et al., 2016; Fassnacht, 2021).

Snowpack properties like snow depth, snow cover, and snow surface wetness are increasingly being surveyed using remote sensing techniques like airborne LiDAR, multispectral sensors, and synthetic aperture radar (SAR) (Currier et al., 2019; Painter et al., 2016; Skiles et al., 2018; Tarricone et al., 2023). These products often work best in-tandem with one another to provide

validation, introducing a strong argument for using multiple methods in assessing snowmelt. C-band SAR has been shown as capable of detecting snowmelt and complements snow cover products from Sentinel-2 (Guiot et al., 2023). However, the resolution of these products may be limited and unable to capture small scale variability as it is influenced by terrain (Fassnacht et al., 2018). One example of higher resolution data are those produced by the Airborne Snow Observatory (ASO) such as spectral albedo, SWE, and depth for basins using LiDAR and multispectral remote sensing platforms (Painter et al., 2016).

These products are appropriate for understanding largescale spatial patterns and resolutions as fine as 3 m; however, these data must rely on modelled snow densities to produce extensive SWE estimates and only represent a brief snapshot in time (Raleigh and Small, 2017). The use of ground-based survey techniques such as ground penetrating radar (GPR) allow surveys at intermediate spatial scales (between point-based stations and airborne platforms) that enable the interpretation of snow properties as they relate to various physiographic controls due to the sensitivity of the radar signal to snowpack properties

(Webb, 2017; Mcgrath et al., 2019; Tarricone et al., 2023; Marshall and Koh, 2008; Bonnell et al., 2021; Mcgrath et al., 2022). Snowpack properties are sensitive to energy balance dynamics, which is typically expressed in four phases: 1) the accumulation phase, 2) the warming phase in which the average snowpack temperature increases towards 0 °C, 3) the ripening phase in which phase changing occurs, but liquid water is retained in the snowpack, and 4) the output phase where further inputs of energy cause melting to leave the snowpack as snowmelt output (Dingman, 2015).  Terrain features like aspect can drastically





alter the energy balance, especially in mid-latitude regions where sun incidence angle will preferentially expose south aspects to shortwave radiation during the day (Molotch and Meromy, 2014; Hinckley et al., 2014; Erickson et al., 2005). Canopy is another terrain feature that can alter snowpack energy balance where subalpine forests reduce accumulation through canopy interception and sublimation (Musselman et al., 2008; Webb, 2017). Conversely, canopies can prolong melt by shielding snow from shortwave radiation (Musselman et al., 2012; Varhola et al., 2010; Lundquist et al., 2013). Canopy will also influence

wind redistribution of snow, increasing the variability of snow accumulation and melt (Mcgrath et al., 2019; Webb et al., 2020b). Similarly, topography can influence wind sheltering and redistribution (Elder et al., 1991; Marks et al., 2002; Winstral et al., 2002). Once the snowpack melts, hillslope processes and soil texture will influence the hydrologic flow paths that form (Webb et al., 2018a; Hinckley et al., 2014; Jencso and Mcglynn, 2011).

Properties such as snow density are often assumed to be uniform across landscapes based off relatively uniform storm

accumulation which can be predicted by air temperature (Valt et al., 2018). Snow density is commonly measured by massing a known volume of a cylinder or a triangular prism, which can be completed as a snow course survey with a federal tube sampler, or with other tools in a snow pit. Additionally, dry snow density can be derived from permittivity (Kovacs et al., 1995; Webb et al., 2021b), which measures the resistance of a medium to the formation of an electric field. Permittivity defines the velocity that a GPR wave will travel through a medium such as snow. These properties allow active radar systems to

measure snow density. Calibrating a snow density model to a specific basin can provide improvement of SWE estimations and has been argued as an important consideration for analyses (Raleigh and Small, 2017; Sexstone and Fassnacht, 2014). GPR provides an opportunity to survey spatial relationships with high precision and control over survey location. Additionally, emerging technologies such as L-Band InSAR depend on knowledge concerning the variability of snowpack properties to constrain uncertainty and improve snow products (Tarricone et al., 2023).

To assess changes in snow density with aspect and position on a slope, we employ L-Band GPR technology. GPR is a broadly used geophysical imaging technology that uses radar wave reflection patterns to determine media properties like dielectric permittivity ($k_s$) (Marshall et al., 2005; Webb, 2017). When paired with precise measurements of snow depth (Clark et al.), $k_s$ can be used to calculate snowpack properties such as snow density (Sommerfeld and Rocchio) (Kovacs et al., 1995; Webb et al., 2018c; Bonnell et al., 2021; Mcgrath et al., 2022). GPR can gather density data with minimal disturbance to the snowpack,

unlike snow pits, and is less time consuming, as it is hauled as fast as a surveyor can traverse the snow. Additionally, $d_s$ data can easily be gathered using a depth probe. We use these techniques to answer the following research question: How does snow density and SWE distribution change throughout the snow season based on aspect and relative location on a hillslope?

## 2 Methods

### 2.1 Study Site Description

The study site for this research is in the Dry Lake watershed, a small watershed that is ideal for studying snow processes in northern Colorado, USA. The watershed is ~0.25 km$^2$ with year-round, hourly data collection from a SNOTEL station and a



remote automated weather station (RAWS) located within the extents of the watershed, respectively (Delong et al.). Elevations range from 2500 to 2660 masl and the primary study area depicted in Figure 1 having a mean elevation of 2545 masl. The SNOTEL station at the site measures a median peak SWE of approximately 510 mm occurring in early April (median date of 100    10-Apr).

The soils in the Dry Lake watershed are primarily loams with very cobbly loam on the south aspect, cobbly sandy loam on the north aspect, and loam on the flatter aspects with observations of highly organic soils in the flat area at the base of the north aspect hillslope (Webb et al., 2018a). Depth to bedrock ranges from 0.12 m to greater than 1 m with a mean depth to bedrock of 0.40 m. A small stream runs from the northeast to the southwest, with an outlet near the SNOTEL station. The lower area 105    consists of forested conifer that is populated with ferns in the summer months and the lower portion of the south aspect is populated by deciduous aspen canopy.

LiDAR data were used to develop terrain and canopy height datasets to quantify the spatial variability of the site (Co, 2016). Using a point cloud filtered for ground surface returns, a 1-meter digital elevation model (DEM) was developed for the site. From the DEM, the north aspect consists of a mixture of north to west facing surfaces and the south aspect consists of primarily 110    south to southeast facing surfaces (Fig. 1b). The north aspect has medium to low solar radiation from terrain shading and the highest solar radiation is seen on the south aspect hillslope (Fig. 1c). Also from the DEM, the north aspect is slightly steeper than the south aspect, particularly at the top of the north aspect survey transect. A second raster was developed from the point cloud which filtered for 1st returns. By differencing the 1-meter DEMs of 1st returns and ground surface returns, canopy height was calculated (Fig. 1d). It is canopied at the base of the hillslopes, with a shorter sparse canopy at the middle of the north 115    aspect, and open canopy near the top of the north aspect. There is less canopy influence during winter months on the south aspect due to fewer trees and those trees being deciduous species. Relative to the north and south aspects, the flat terrain shares low angle north to west facing surfaces and contains the tallest canopy height resulting in moderate solar radiation is moderate. These spatially variable physiographic parameters are important to consider when dealing with seasonal snowpacks, where the energy balance is sensitive to parameters like terrain and canopy cover shading.





**Figure 1: a)** The location and imagery of the Dry Lake watershed including general location within the western USA, where study area within the watershed where transects were established, and the locations of the RAWS and SNOTEL stations. (Imagery gathered via Google Earth Pro v. 7; Google Earth, 2024; © Google). **b)** Aspect map, **c)** solar radiation model, **d)** percent slope of terrain, and **e)** canopy height using LiDAR data. Survey transect locations are indicated by black circles.

120



## 2.2 Data Collection

In the winter and spring of 2023, seven transects were established to collect data at varying positions on the north and south aspects as well as the flat terrain. The spatial distribution of these transects were designed to capture changes in snow properties related to aspect and position on slope including the base, middle, and top of slopes (Fig. 2). The flat terrain transect was taken by traversing a circle around the SNOTEL station whereas all other transects were ~20 m in length perpendicular to the fall line (i.e., parallel to slope contours, Fig. 2). These data were collected approximately once every month from January to May, resulting in 5 survey dates. All transects included GPR data collected with surface-coupled, common offset GPR units pulled over the snow surface. The first three surveys used a plastic sled hold the GPR, and the GPR was pulled freely without a sled during the final two surveys. Both systems were manually towed behind an individual on skis. Two systems were used: a pulseEKKO GPR system and a Mala Geosciences GPR system. The pulseEKKO system used a shielded antenna at 1000 MHz. The Mala GPR system used 1600-MHz and 800-MHz antennas and was only used during a single late February survey. Following the GPR, depth measurements were collected in the track of the GPR at 2-meter spacing (Webb & Mooney, 2024a). Snow pits were additionally dug to measure bulk density of the snowpack within the flat terrain, on the north aspect, and at the base of the south aspect (Webb & Mooney, 2024b). GPR transects were conducted next to the pit to calibrate GPR-derived density measurements for each survey date. When time allowed, 1000 cm$^3$ wedge cutters were used to determine a density profile at 10 cm intervals. During days when time was limited, a profile of ~50 cm long cores with a diameter of ~6 cm were used to estimate snow density. Thus, each snow pit had 2-20 measurements of density, depending on the time available for pit observations to derive bulk density. Notes were also taken about observations of liquid water pooling or ice lenses with depth of occurrence and thickness.

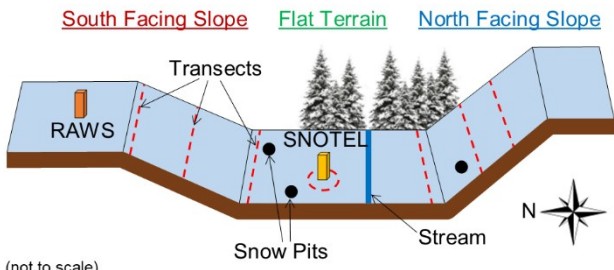

**Figure 2: Locations of GPR and depth probe transects on the hillslopes of the study area (not to scale).**

## 2.3 Data Processing

Radar data were processed using ReflexW for each transect. The first processing step was to apply a dewow filter, which removes low frequency noise in the time domain by subtracting a running mean from the central point. Applying this filter allows the trace to have a mean of zero which removes any slope in the trace and allows for positive and negative signals throughout the trace. A time-zero correction was applied next by selecting the air wave first break. A gain filter was then applied to account for signal attenuation and geometrical spreading loss as the wave propagates through the snow by amplifying





the strength of later arrivals. An AGC-Gain function was used which applies a multiplying factor to successive regions of the trace in time, dampening un-necessary noise. The next step was to edit trace range along the x-axis. This step can be used to remove time periods when the GPR was not moving. During data collection, there are periods of standstill between when the device is powered on and when the transect data are being collected, and between when the transect ends and the device is turned off. Removing the traces before and after effectively crops the radargram to only include the transect data and not oversample the ends of the transect. Finally, a background removal filter is applied. This filter removes any excess noise and excess banding that may be present in the traces. In this step, the processing is set for all data at 1 ns or greater to retain the surface wave, which retains the clarity of the surface wave and soil-snow interface wave during picking. Next, the surface and soil-snow interface reflections were 'picked' using a semi-automatic picking tool in ReflexW. Figure 3a displays an example of a radargram showing the snow surface reflection and snow-soil interface reflection. The surface wave reflection was then subtracted from the snow-soil interface reflection to determine the two-way-travel time (TWT) through the snow (Webb & Mooney, 2024c).

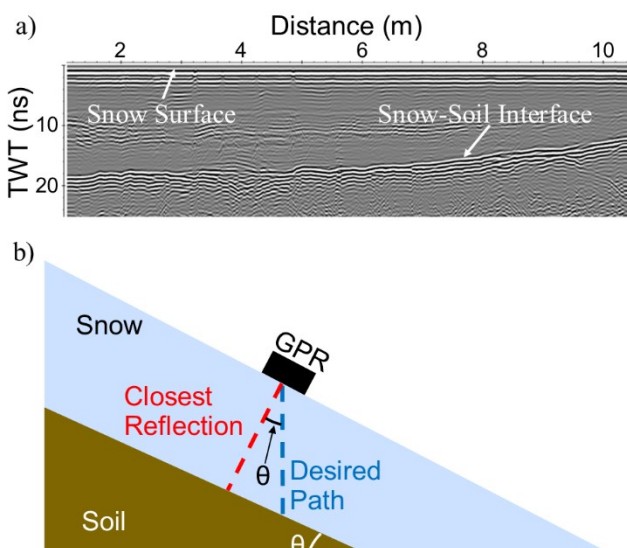

**Figure 3: a) An example of a processed radargram and the snow surface and snow-soil interface reflections. b) A graphical depiction of the correction for slope angle to align TWT and depth probing for each transect.**

The median TWT (ns) for each GPR transect and associated average measured $d_s$ (m) was used for the following calculations to estimate bulk snowpack density:

$$v = \frac{d_s}{\frac{TWT}{2}}$$

where $v$ is the radar wave velocity in m ns$^{-1}$, and $k_s$ is calculated with the speed of light ($c$) in a vacuum:





$$k_s = (\frac{c}{v})^2$$

and bulk density ($\rho_s$, kg m$^{-3}$) is estimated using *Kovacs et al.* (1995):

$$\rho_s = \frac{\sqrt{k_s}-1}{0.845} * 1000$$

175   SWE was also calculated by multiplying the estimate of $\rho_s$ by the observed $d_s$.

When traveling in sloped terrain, the GPR TWT needs to be corrected since a GPR will receive the reflection of the closest reflector that will tend to be normal to the slope. Thus, we adjusted the TWT to be in-line with gravity to ensure the same direction of depth probing by dividing by the cosine of the slope angle (Fig. 3b).

## 2.4 Meteorological Data

180   Hourly data from SNOTEL and RAWS stations in the Dry Lake study site were utilized for the 2023 water year. These data are used to contextualize field measurements taken during the observation period as inputs into a physical snowpack model. Downward longwave radiation was collected for the area using Hydrology Data Rods, NLDAS Primary Forcing Data (Teng et al., 2016; Xia et al., 2012).

The Dry Lake Colorado RAWS station is at 2536 m (8320 ft) elevation on the ridge of the south aspect of the study area while

185   the Dry Lake SNOTEL station is centrally located in the watershed in flat terrain at 2521 m (8271 ft). The RAWS data records include hourly precipitation, wind speed, wind direction, relative humidity, max wind gust speed and direction, and incoming shortwave radiation. The SNOTEL site data include hourly measurements of precipitation, SWE, wind speed, air temperature, and snow depth. Midnight values are quality controlled by snow survey staff to account for error in sensors; however, hourly data is not edited at the time of this study. Using the following rules, hourly data from SNOTEL was corrected to create

190   continuous, hourly data for model input: 1) Accumulated precipitation cannot decrease, 2) If there is an increase in snow depth, there must be an increase in SWE, 3) An increase in SWE should prompt an increase in accumulated precipitation, and 4) Hourly data must fit within the limits of the preceding and following midnight values, but hourly patterns can be preserved. From these hourly data, $\rho_s$ was calculated for the SNOTEL station by dividing SNOTEL observed SWE by $d_s$. Physically impossible densities were removed (i.e., negative densities and those greater than the density of water) by replacing those

195   values with the value from previous timestep value. Figure 4 displays the processed SNOTEL SWE, cumulative precipitation, $d_s$, and $\rho_s$ data used for this study.



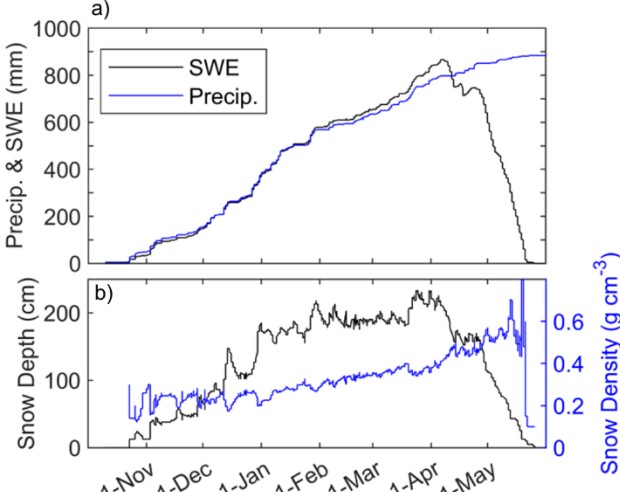

**Figure 4: SNOTEL data for the 2024 water year showing a) observed SWE and cumulative precipitation, and b) observed snow depth and calculated snow density.**

## 2.4 SNOWPACK Modelling

The SNOWPACK model (Bartelt and Lehning, 2002) simulates seasonal snowpack based on weather station data. This study uses SNOWPACK to represent energy balance changes occurring on each aspect of the watershed to contextualize observations made in the field, with the primary objective of informing the researchers about the timing of snowmelt events. SNOWPACK discretizes the vertical snow profile into layers, increasing through accumulation and decreasing through melt and compaction. In addition to closing the mass and energy balances per time step, the model includes physically-based routines for internal snowpack processes (e.g. liquid water transport and energy exchange) and a unique empirical scheme for snow grain metamorphism. Simulated snow depth, SWE, snowpack temperature, and stratigraphy have been extensively validated for SNOWPACK (Jennings et al., 2018a; Lundy et al., 2001; Meromy et al., 2015; Rutter et al., 2009). SNOWPACK has also been shown as a successful tool in predicting snow liquid water content (LWC) in previous studies (Webb et al., 2018c; Webb et al., 2020a; Webb et al., 2021a).

Simulations were run at hourly time steps with quality-controlled observations of air temperature, relative humidity, wind speed, incoming shortwave radiation, incoming longwave radiation, precipitation, and ground surface temperature to simulate the accumulation and melt of a snowpack. The precipitation phase threshold was increased from the default SNOWPACK value of 1.3°C to 2.5°C because the Rocky Mountains of the western United States have some of the highest rain-snow thresholds in the northern Hemisphere (Jennings et al., 2018b). Turbulent energy exchange was simulated using the bulk Richardson number approach as this stability correction produced the best model performance at another subalpine site in Colorado (Jennings et al., 2018a). SNOWPACK simulates the transport of liquid water using Richard's equation (Wever et al., 2014).



The first model is a south facing exposed ridge at the top extent of elevation for the study area, using mostly RAWS station
data (which is positioned on the ridge of the south aspect). The second model represents the flat terrain of the study area using mostly SNOTEL data. The third location modelled is on the north aspect. The north aspect and south aspect models do not consider canopy effects to represent general hillslope conditions.

## 3. Results

### 3.1 Transect Data

We found transects in the flat terrain transects show snow depth and SWE increasing during the accumulation period similar to SNOTEL data, January through April. SNOTEL peak SWE occurs on April 6 at 866 mm, followed by rapid decreases in snow depth and SWE. In this region, transect data showed similar peak snow depth, but slightly lower SWE and $\rho_s$ values during the 1-Apr survey (Fig. 5d). In general, the flat terrain transect data compared well with SNOTEL data.

Snow depth on the north aspect follows a similar pattern to the flat terrain with increases during the accumulation phase and a
rapid decrease starting in April, though this period also resulted in large increases in $\rho_s$ indicating an increased rate of densification while SWE increases slightly. North aspect $d_s$ values are highest overall with the top of slope consistently producing the deepest snow throughout the season (Fig. 5g). However, we observed two distinct patterns in $\rho_s$ on the north aspect. The first pattern is for the top and middle of the north aspect showing a relatively consistent $\rho_s$ through the early surveys, and a large increase of $\rho_s$ for the May survey (Fig. 5f-g). The second pattern occurred at the base of slope showing a consistent
increase from February to April. This base of the north aspect also resulted in an unrealistic value during the May survey that we interpret as the result of excessive liquid water content due to a very low radar velocity and high relative dielectric permittivity (Bradford et al., 2009; Webb et al., 2018c). The SWE estimates from transect data follows these same $\rho_s$ patterns on the north aspect.

The south aspect had different patterns relative to the flat terrain and north aspect (Fig. 5). The $d_s$ at the top and middle position
of the south aspect show gradual increases from January to April, with both gains and losses in SWE during this time (Fig. 4a-b). The base of the south aspect sees a similar pattern of increasing depth, but with SWE consistently increasing from January through April surveys (Fig. 5c). All transects on the south aspect experienced a decline in SWE from the April to May surveys, though the smallest change occurs at the base of the south aspect, coinciding with a large increase in $\rho_s$ at this location (Fig. 5a-c).





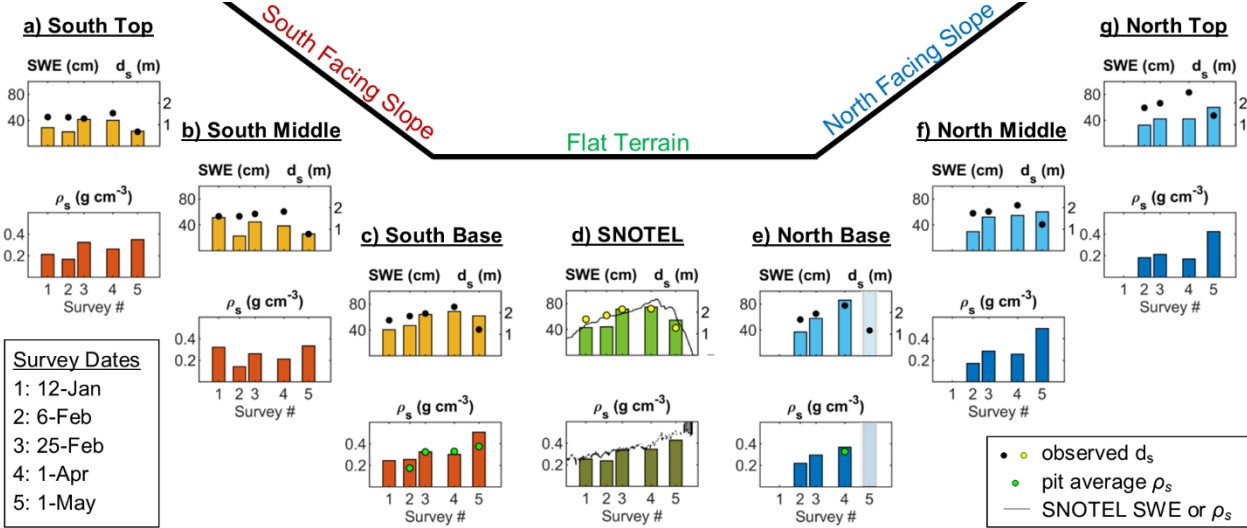


**Figure 5: Results from the transect observations including calculated SWE, observed $d_s$, and GPR derived $\rho_s$ for: a) South Top, b) South Middle, c) South Base, d) Flat Terrain around the SNOTEL station, e) North Base, f) North Middle, and g) North Top. Pit measured average densities are shown when collected, and SNOTEL station data are displayed for additional comparisons of SWE and $\rho_s$. Note that the GPR results that gave unrealistic values due to the presence of liquid water is slightly greyed in panel e.**


## 3.2 Snow Pit Observations

In general, the snow pits show patterns of increasing density with depth and time, as expected, with ice lenses and layers forming from the upper to mid snowpack in all pit locations (Fig. 6). Pits dug at the base of the south aspect showed a single ice layer during the 28-Febrary and 1-April surveys. This ice layer was approximately 4 cm thick at ~150 cm above ground in February and approximately 11 cm thick ~70 cm above ground in April (Fig. 6a). The flat terrain pit did not have any ice lenses/layers in January, but one ice lens was observed in April that was approximately 3 cm thick and ~230 cm above the ground (Fig. 6b). The north aspect only had a single snow pit observation during the April survey, but ten ice lenses/layers were observed throughout the snowpack from 30 cm to 210 cm above the ground, all were approximately 1-2 cm thick (Fig. 6c).



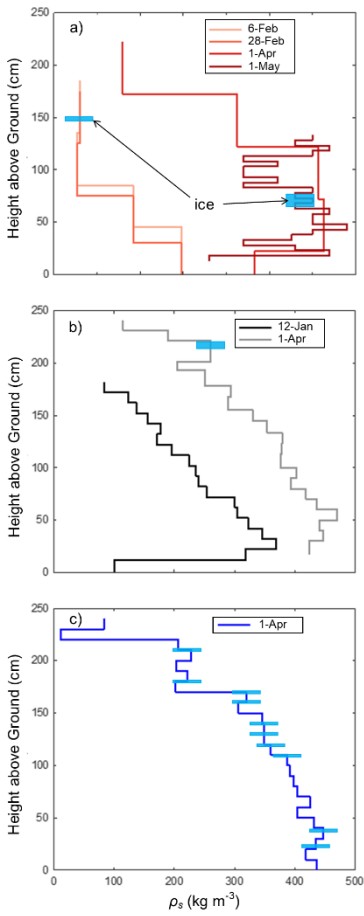


**Figure 6: Snow pit observations of $\rho_s$ and ice layers/lenses for: a) the base of the south aspect, b) flat terrain, and c) north aspect.**

### 3.3 SNOWPACK Modelling Results

Modelling of $d_s$, $\rho_s$, and SWE were completed using the SNOWPACK model to simulate accumulation and melt on the north

aspect, south aspect, and flat terrain areas of the study site. The north aspect model indicates the largest snow depths and longest snow persistence, as expected due to terrain shading (Fig. 7a-b). The flat terrain model produces a model matching the SNOTEL data well during accumulation, but with slightly different melt rates in May (Fig. 7a-b). The south aspect model shows the lowest snow depth and the earliest melt out date. (Fig. 7a-b). The $\rho_s$ modelled in SNOWPACK is similar across each aspect until April when melt begins. All model simulations indicate a spike in density prior to completely melting out, but with

different amplitudes and timing. The modelled SWE shows similar patterns relative to SNOTEL data. The north aspect model accumulates more snow than the SNOTEL data, as expected. Peak SWE in the north aspect model occurred on April 24 at ~920 mm, whereas SWE peaks in both the flat and south aspect models on April 5 (~810 mm and ~285 mm, respectively), the date of a snowstorm prior to a period of warmer weather.



All model simulations show intermittent surface melt events (Fig. 7c-d), with the largest and most regular occurring on the

south aspect simulation (Fig. 7e). Simulated bulk volumetric LWC on the south aspect increases to 1% or more nine times

from December through March (Fig. 7e). The other north aspect and flat terrain model simulations do not see volumetric LWC

values greater than 0.5% for that same period of December through March (Fig. 7c-d).

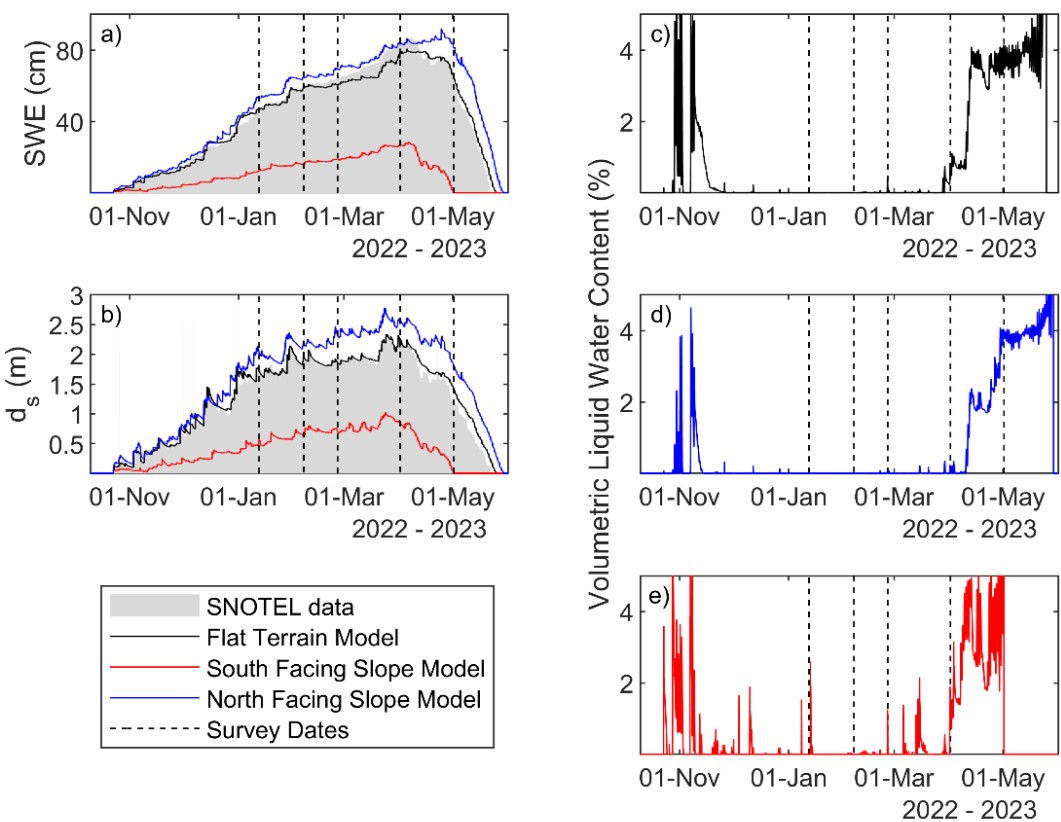

**Figure 7: Results from the SNOWPACK model simulations including a) SWE, b) $d_s$, and c) – e) volumetric liquid water**

**content. Results show comparison to SNOTEL data as well as timing with survey dates.**

## 4. Discussion

This study observed snow density variation with aspect and position on slope using pit calibrated GPR transects. The results

show mid-season melt occurring on the south aspect that redistributes SWE down towards the base of the slope, and pooling

of LWC in the snowpack at the base of the north aspect during the ripening and melt phase. Variation in snow depth and



density along both hillslopes have implications for SWE distribution and peak timing, indicating the importance of aspect-specific considerations for modelling of SWE and melt processes (Sexstone and Fassnacht, 2014; Lopez-Moreno et al., 2013). The flat terrain area did not produce any unusual results. GPR values matched well with SNOTEL $d_s$ and SWE measurements (Fig. 5). The flat terrain SNOWPACK model also showed results that matched well with observational data (Fig. 7). SWE varied slightly from the measured to the modelled data, likely due to precipitation uncertainties relative to snow on the ground

as observed by the snow pillow.

Northern hemisphere incidence angle of the sun allows for more exposure on the south aspect compared to the north aspect, which is shaded more of the time. This influences the energy balance of the snowpack by reducing energy inputs to the north aspect and increasing energy inputs to the south aspect, resulting in differences in accumulation and melt dynamics (Molotch and Meromy, 2014; Erickson et al., 2005). Additionally, the south aspect doesn't receive canopy shading in the winter because

it is canopy free in the top half of slope and is populated with deciduous Aspen on the lower half of slope, which lose their canopy during the winter (Musselman et al., 2008; Varhola et al., 2010). This difference in canopy cover is likely attributed to aspect as the deeper snow and increased soil moisture on the northern aspect increases the amount of plant available water for vegetation growth (Webb et al., 2023). This increased exposure on the south aspect results in SWE losses from three mechanisms: melt, sublimation, and wind scouring. Wind sensors on the RAWS station indicate that windspeeds top out at 6-

10 m/s with most gusts traveling northeast. The precipitation at this ridgeline sensor is lower compared to the SNOTEL sensors in the flats, likely indicating strong winds blowing snow over the gauge. These kinds of wind could contribute to scouring of snow. Blowing snow is also more susceptible to sublimation (Vionnet et al., 2013). Modelling of snow depth and SWE on the south aspect are largely a product of precipitation input from RAWS data, resulting in lower values compared to measurements (Fig. 7). Melt out dates reflect these lower precipitation inputs as well, with observable snow depth surveyed on May 1st while

the model simulated this as the last day of snow cover for the south aspect (Fig. 7). Despite model weaknesses, the LWC parameter shows when surface melt occurred due to its root in physical processes. The south aspect model reveals several mid-season surface melt events that are not present in the flat or north aspect models, which is likely a response to increased solar radiation exposure that were also qualitatively observed during surveys. These mid-winter melt events on the south aspect coincide with increased density at the base of slope, indicating a likely downhill migration of SWE through intra-snowpack

flowpaths (Webb et al., 2020a; Webb et al., 2022). There is also the observation of an ice layer at the base of the south aspect that is indicative of lateral flow in sloping terrain (Webb et al., 2018b). Observations of surface melt occurring on the south aspect also included small runnels forming late in the afternoon during the April survey, which is further supporting this interpretation. Additionally, soil moisture sensors at the SNOTEL station indicate a steady rise in soil moisture that align with snowpack accumulation, indicating a steady source of moisture throughout the winter (Fig. 8). With lateral groundwater fluxes

from outside this watershed assumed to be minimal, the source of soil moisture rise is likely from melting snow on the south aspect as snow elsewhere in the watershed remains cold enough to not provide moisture inputs. These results indicate the input of snowmelt from the south aspect providing connectivity to the stream and water sources to potentially maintain baseflow through the winter, though streamflow data are not available for this location and requires further research in the future. Further

quantification of subsurface properties such as porosity and saturation of soils on the slope could clarify these processes and fully describe vadose zone hydrologic connectivity and water movement.

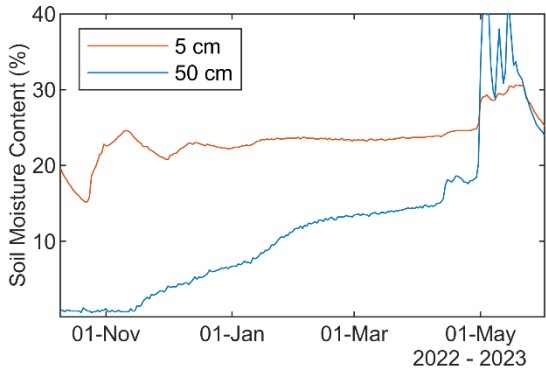

**Figure 8: Soil moisture data from the SNOTEL site at depths of 5 cm and 50 cm.**

The north aspect is an area of lower solar radiation exposure compared to the south aspect. This is evidenced by greater snow depths and later melt throughout the winter and spring seasons. The region is partially forested as well, which is likely a result of terrain shading and greater water availability during the growing season . This coniferous canopy remains intact throughout the winter months, providing shelter from wind and solar radiation, allowing snow to accumulate and persist longer. The survey transect higher up on the north aspect resulted in greater depths as a result of low canopy interception. Further down at the middle of the slope, depth decreases slightly with minimal SWE differences relative to the top of the slope. The transect in the middle of the north aspect has partial canopy coverage with parts near the drip edge of trees that likely resulted in some interception but also canopy sloughing that caused the lower depths and higher densities at this location relative to the top of the north aspect. The base of the north aspect is in a small opening of the mostly forested location of this study, though interception did not cause a large difference in accumulated snow depth Fig. 5e). The most notable difference at the base of the north aspect is the steady increase in snow density through the observation period, with an unrealistic increase in density during the May survey. Once the snowpack begins to ripen, density spikes to values that are not physically possible, which is an indication of GPR signal slowing from liquid water in the snowpack. This could be a result of the exposed areas of the slope producing meltwater which flows downhill and pools at the base of the slope as previously observed at this site (Webb et al., 2018a). Unlike the south aspect, most of the SWE has remained on the hill, rather than melting intermittently with mid-season melt events. This excess of water, paired with fine-grained soils with low infiltration capability, could explain pooling of liquid water at the base occurring with the onset of the melt phase. Snow pits dug on April 1st at the base of slope further support this interpretation, as several ice lenses/layers distributed throughout the snowpack were observed indicating multiple hydraulic barriers with the potential to divert liquid water laterally in the snowpack the entire length of the hill slope (Webb et al., 2018b).



The energy balance proved to have a large effect on field data and modelling as the south aspect model encompassed greater
energy inputs and exposure than the north-aspect, resulting in different accumulation and melt dynamics. While these specific
basin dynamics are not applicable to every snowpack, there are some general patterns that can be applied to areas with similar
characteristics. For instance: 1) Open canopy, south aspects have greater potential for mid-winter melt events, causing a
redistribution of SWE downslope to increase SWE and soil moisture (Fig. 9a); and 2) north aspects may experience lateral
flow of water through snow and in the shallow subsurface causing accumulation and pooling of liquid water at the base of
slope during spring ripening and snowmelt (Fig. 9b; Webb et al., 2018a). Figure 9 offers an update to the *Webb et al.* (2018a)
conceptual model of aspect controls on liquid water movement, with descriptions of the dominant processes during the winter
and spring periods.

The main objective of this study was to determine how snow density changes with aspect and position on hillslope. We found
that sloped areas can have quite different melt dynamics which can greatly influence snow density. In particular, the base of
slope seemed to be an area of greater SWE following different melt mechanisms (Fig. 9). Each aspect melts at different times
because of varying energy balance dynamics. The south aspect is responding to mid-winter melt, which is distributing mass to
the base of slope during the middle of the winter whereas the north aspect is experiencing primarily accumulation during winter
and distribution of mass through melt processes during the spring ripening and snowmelt periods in April.

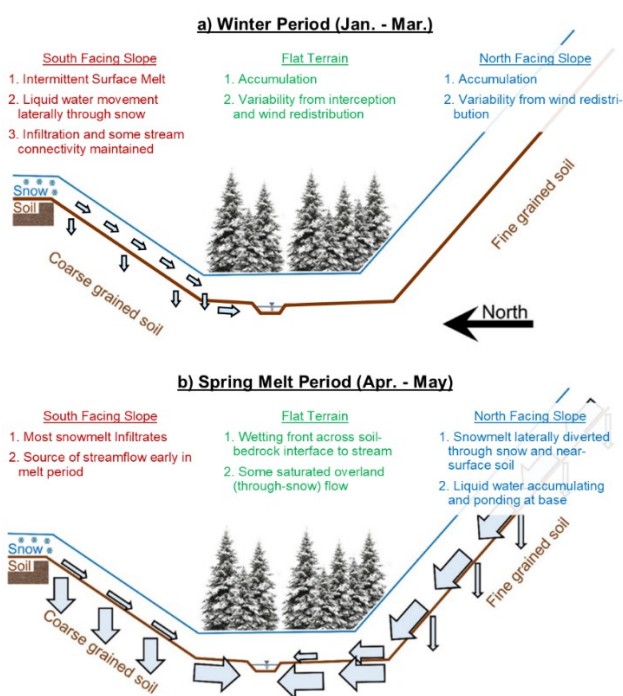


**Figure 9: Summary of processes during a) the winter period (January through March) and b) spring melt period (April
through May). Panel (b) is modified from Webb et al. (2018).**





Assessing patterns in snow density and the influence of the movement of liquid water throughout a watershed and snow season
can provide important context to measuring and modelling snow (Webb et al., 2022). Snowmelt and catchment liquid water
input into a system have historically been associated with snowmelt rates; however, snowmelt rates are dependent on complex
energy balance interactions between the snowpack and its environment. The traditional 4-phase snowpack model (Dingman,
2015) may not be representative for all snowpacks everywhere in a single watershed at a given time, especially when
considering hillslope processes. Position on slope, aspect, and snowpack phases were found to be factors in predicting snow
density and presence of liquid water. Areas with higher energy input may see a greater range of density and more dynamic
snowpack conditions. Paired with well-known depth variation, these parameters could have a compounding effect on SWE,
further emphasizing the importance of quantifying spatial variability of density at the catchment scale. These results support
further quantification of catchment scale density for measurement of SWE, especially on different aspects as they have a
significant influence on snowpack energy balance. Similar studies are needed to understand density variation in systems with
different energy balance dynamics, or conversely, future projections of energy balance scenarios.

## 5. Conclusions

This study found that aspect produces snowpack melt and SWE distribution dynamics that are different from a traditional flat
area conceptual model. In general, there is a pattern of downhill SWE migration and densification at the base of either hillslope
which is largely influenced by energy input timing. Of these, the south aspect was found to be susceptible to mid-season melt
events which increased snow density through the redistribution of SWE via the lateral flow of liquid water to the base of the
hillslope. The north aspect behaved more like the flat areas during the accumulation phase, with a large change at the onset of
April melt causing liquid water pooling at the base of the hillslope. These differences between aspects are most related to solar
radiation inputs and preferential terrain shading.

## Data Availability

SNOTEL data are available from the online repository (https://wcc.sc.egov.usda.gov/nwcc/site?sitenum=457) and RAWS data
are available through the DRI data repository (https://raws.dri.edu/cgi-bin/rawMAIN.pl?coCDRY). Field survey data are
available in the Dry Lake Watershed collection in CUAHSI Hydroshare (Webb, 2024;
http://www.hydroshare.org/resource/4aff38a0cbb24456be4e99987e808abb).





**Competing Interests**

The contact author has declared that none of the authors has any competing interests.

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
