# Peer review of "Aspect Controls on the Spatial Re-Distribution of Snow Water Equivalence through the Lateral Flow of Liquid Water in a Subalpine Catchment"

_EGUsphere, 2024_

## Author Comment (AC1)

In the study 'Apsect Controls on the Spatial Re-Distribution of Snow Water Equivalence in a Subalpine Catchment', the authors investigated how slope aspect influences snow accumulation and melt dynamics using ground penetrating radar (GPR) transects and snow pit and snow core density measurements during the water year 2022-2023 in a small catchment in Colorado, USA. The authors also used the SNOWPACK snow model to support their observations and further develop a conceptual model of water redistribution in the snowpack based on aspect. Overall, this study provides interesting results using different instruments and tools (GPR surveys, snow pits, weather stations and a multi-layer snow model). I acknowledge the authors' efforts in combining these elements to propose a conceptual framework to better understand how aspect may exert a control on the evolution and (re)distribution of snow water equivalent (SWE). However, the proposed conceptual model is based on indirect observations and modelling results rather than on field evidence, which should be clearly stated. I also believe that the manuscript would benefit from a more straightforward storyline and some elements such as the influence of forest canopy on SWE should be detailed. After addressing these issues and some additional minor and technical comments, I am confident that this paper will be a good and relevant scientific contribution to snow research and will provide insights for future studies.

Thank you for the comments and very detailed review. We appreciate the constructive comments that we agree will improve the interpretation of this work. Below are replies to specific comments in blue as well.

Major comments.

Structure of the introduction

The introduction needs to be clarified so that the reader understands the relevance of the study. Here are some suggestions:

The first two paragraphs would benefit from being restructured into one and generally rephrased and shortened.

Agree. This will be done for brevity and clarity.

The third paragraph (l.47 to l.73) is too long and goes in all directions. I would recommend the author to split this paragraph into two, based on the description of methods (l. 47 to 60) and landscape control of snowpack properties (l. 61 to 73).

Agree, this will be revised as suggested.

The last paragraph should be completely revised. Lines 85 to 91 should be moved earlier in the Introduction. I would recommend that authors make links to earlier parts of the introduction to emphasise the relevance of their work. I also strongly suggest that a research aim for this work be clearly defined (which is not the case at present).

We will revise this paragraph to more clearly define the research aim that is currently in the last sentence as the research question only. We will revise to state the broader aim early in the paragraph and re-phrase the sentences to more clearly illustrate how we are approaching the study.

Modeling setup

Several questions remain about the modelling setup.

What parameters did the authors use for each simulation (north, south flat)?

We will add supplementary material that has all of the input information for every parameter used.

Were soil layers defined in these simulations? If not, is there a reason for this, since the authors have a description of the soil (see lines 101 to 104)?

We did not define soil layers as the main purpose of the modeling in this study is to inform us on the timing of snowmelt events. (line 203)

There are several parameters in SNOWPACK that are site-dependent or have to be chosen arbitrarily by the modellers. What values did the authors use in their simulations for these parameters?

These will be included in the supplementary material mentioned above.

I understand that the canopy was taken into account for the flatland simulations. This implies that several other parameters have to be specified. What values did the authors choose?

We apologize for any confusion, canopy was not considered for the flat simulations. This will be clarified in revisions. The flat terrain pit and transect was in a large clearing that had minimal effects from canopy that needed to be simulated.

I think adding an appendix with the different keys enabled in the model and the parameters used for each simulation would be a clever way to answer these questions. Consider also explaining any arbitrary choices and how site-specific values were obtained.

The supplementary material will be made and will include explanations for modeling choices.

Influence of the canopy

One of my main concerns with this article is that the influence of canopy cover on the spatial distribution of SWE is not adequately addressed. While canopy control is presented in the Introduction (l. 66 to 71) and some results are interpreted based on the canopy in the Discussion (l. 294-298; l. 325-334), the role of vegetation is not presented in the Results section. While I respect the authors' decision not to make this the main focus of their paper, I think the article would benefit from consistent treatment of the

influence of canopy on snow redistribution alongside aspect control. Please consider including this in a revised version of your manuscript.

This will be added to the interpretation for clarity. We will point out the specific influences of canopy on it's own, but also the feedback loop between north aspects having more water available for plants and generally producing thicker canopies as a result.

Limitations

The fact that not every interpretation is based on field evidence is not critical. However, I would suggest that the authors include a section on the limitations of their study. This would help the reader to better contextualise some of the analyses, especially with regard to SWE redistribution processes through the snowpack.

We agree. Clarification on what is directly observed and what is not will be made to ensure clarity in the revised manuscript.

Minor comments.

l. 16. Explicitly mention the use of snow pit and soil moisture monitoring measurements in the abstract.

Will do.

l. 32-33. I do not think this sentence is necessary. Please remove.

Will do.

l. 33-36. This sentence is difficult to understand. Please rephrase and break it down into two sentences.

Will revise.

l. 42-46. I am not sure if I understand this sentence correctly or if it is necessary for the general understanding of your study. Please rephrase or clarify this idea.

Will clarify.

l. 57-60. Please consider breaking it down into two sentences.

Will revise as suggested.

l. 61. I think starting a new paragraph here would improve the readability of the introduction.

It is a new paragraph, but the format of the journal and the previous sentence ending at the margin make it appear to not be.

l. 66-68. You mention the energy balance, but then refer more to the mass balance of the canopy (e.g. accumulation by canopy, interception). Perhaps you should just mention that the canopy changes the energy and mass balance of the snowpack.

Good point. Will revise as suggested.

l. 74. Consider specifying the 'bulk' snow density here. The distinction is particularly important as you go on to present detailed density profile measurements (Fig. 6). I would also consider adding a few words on how snow density at the layer scale varies with landscape characteristics.

Will revise as suggested.

l. 77-80. Why is the derivation of snow density from permittivity given for dry snow only? A few words about this method applied to wet snow would be relevant.

Will revise as suggested to further clarify why we did not apply the wet snow form as well.

l. 82. Please clarify the meaning of 'spatial relationships'.

Will revise as suggested.

l. 87. I am a bit uncomfortable with ks being the symbol for the dielectric permittivity of snow. ks often refers to the thermal conductivity of snow. Please consider using the symbol '$\varepsilon$' for permittivity.

Will revise as suggested.

l. 90. Please delete the following: 'being dragged as fast as a surveyor can traverse the snow'.

Will revise as suggested.

l. 90. What is ds? This variable has not yet been defined.

Will define as snow depth when it is first introduced.

l. 100. Please specify the historical period of the measurements.

We will add this information.

l. 103. Do you have the average thickness of the litter? If so, please specify.

Yes, we will add this information. It is generally 10 – 15 cm on the north aspect, and somewhat deeper at the toe of the slope. This if from memory and I will confirm from notes for the revisions.

l. 110-119: Please consider shortening the details of how the DEM and canopy height models were developed.

Will revise as suggested.

l. 114. I understand the meaning of the word 'canopied', but as this term is quite uncommon, it distracts the reader from the text. Consider using another term.

Will revise as suggested.

l. 117. Please check and correct the end of this sentence.

Will revise.

l. 118-119. I do not think this sentence is necessary.

Agree. Will delete.

Figure 1.

Why is north pointing to the left? I think it would be better to rotate your map 90 degrees and make it poiting upward.

It is due to the 3D birds eye view where if North were up some of the transect is actually difficult to see in the image.

I am not sure that the orientation of 1a is the same as 1(b to e), please check. Consider adding the river to figure 1a.

We will double check this and correct if necessary during revisions.

1 b and Fig. 1d could be combined into one figure using elevation lines.

We also looked at this and believe this is a style choice where we prefer the current version.

1c could be removed. If the authors decide to keep it, please indicate how shortwave radiation was calculated.

We will consider both options during revisions.

l. 121-123. Please rephrase. It took me a few reads to understand the sentence.

We will revise the caption to be more clear.

l. 126. Please indicate the exact start and end dates of the data collection.

Will do in the revisions.

l. 131-133: Please revise these two sentences. It seems that some words are missing...

No words seem to be missing, but we will try to revise a bit for clarity.

l. 135. Why did you use two different systems? And how might this affect your results?

One of the systems needed to go to AK for a different campaign to have the same system as other teams. The physics should remain the same for this change in frequency so the only impact would be a change in precision and uncertainty in two-way travel times.

l. 137-142: Please consider adding a table of snow pit measurement dates, indicating which density measurement method (wedge cutter or tube) was used on which date.

Will add as suggested.

l. 142. Please indicate how water ponding and ice lenses were identified. Perhaps a photo of a snow pit experiment (if you have one) would be relevant here.

From my memory hen photos were taken they did not show ice lenses or water ponding very well, unfortunately. We will go back through the photos to check, though.

l. 145. I would remove Figure 2 from the manuscript.

We will consider either removing it, or incorporating the information into Figure 1.

l. 147. Please add a few words about ReflexW.

Will revise as suggested.

l. 147-163. I really appreciate this paragraph, which is fluent and easy to read. I think a conceptual figure of the multi-step data processing method would be nice. Please consider replacing Figure 3 with this conceptual figure.

We believe that a flow chart or conceptual figure of the steps may be a bit redundant given the details in the paragraph.

l. 167-175. I get quite confused with ds and ks. Defining ds first would definitely help, but still. This part with the equations is a bit messy. Please check that the correct variables are used and described. Please also include the number of each equation.

Will double check and revise for clarity and formatting.

l. 180-181. These two sentences should be merged into one.

Will revise as suggested.

l. 182-183. This sentence should follow the description of the data provided by the SNOTEL and RAWS stations.

Will revise as suggested.

l. 191. Please mention that redistribution (e.g. by wind or canopy unloading) is neglected.

Will revise as suggested.

l. 198. 2023 water year? Consider adding a label on the x-axis of the plot instead.

Will revise as suggested.

l. 200. Could you add a sentence explaining why SNOWPACK was used instead of another model?

For our purposes, any snow energy balance model would have worked. It was somewhat due to convenience as this model was taught for different purposes (profile liquid water and temperature imagery) in a class during this master's project coursework. But, SNOWPACK is an option that is often used for studies looking at liquid water content.

l. 204-205. This is not exact. Please be more specific about how SNOWPACK creates, removes or merges snow layers.

Will revise as suggested.

l. 206. A clearer explanation of the liquid transport processes could be given here. See Wever et al. (2014 - https://doi.org/10.5194/tc-8-257-2014).

Will add more information on the use of Richards equation in SNOWPACK.

l. 207. In fact, SNOWPACK relies on fundamental physical principles to simulate snow metamorphism. Please remove the statement that it has 'a unique empirical scheme'.

Will revise as suggested.

l. 235-237. Can this be confirmed by any snow pit observations?

We did make snow pit observations during the May survey and will add further text concerning observations.

Figure 5. While I appreciate the effort put into this figure, I think it could be simplified. The way the figure is presented makes it difficult to compare results from different sites. Also, in section 3.1 of the the text, the frames (a, b, c ...) are not presented in any order, which makes it confusing. I would suggest a typical side-by-side plot where we can more easily compare the north-facing slope, the south-facing slope and the flat terrain.

We will present the frames in the text in a more logical order, and create a figure with panels side by side. That should not be too difficult to create for comparison and see which one is preferred in the revised version. We agree that the current version is difficult to quantitatively compare values between sites for specific dates.

l. 251. As snow pit observations were not systematically performed during your field surveys, I would recommend listing each snow pit date in a table (perhaps in the method section).

Will revise as suggested.

l. 257-259. That is an interesting observation. Could you elaborate?

We can elaborate a little bit more about the depths of the ice lens observations and thicknesses and how there was a larger density across certain depths. Is that what you are looking for?

Figure 6: Please increase the size of the axis labels. Consider also using a colour gradient to display density profiles (see Fig. 3c-d from Bouchard et al. (2022 - https://doi.org/10.1002/hyp.14681) as an example). This would allow each profile to be shown on the same frame and would make them easier to compare.

Will revise axes labels as suggested and look further into the color gradient for density profiles.

As a general comment, be sure to follow a same order of presentation of the results (e.g. 1. flat, 2. south, 3. north) in the different sections where you refer to them.

Will revise for this order of presentation.

l. 263. Although this is not the objective of the study, I think it would be interesting to compare the simulation results for snow density with your snow pit observations. This would give a better idea of how the model performs at your site. Consider adding this analysis.

We will add this information.

l. 272. The difference in peak SWE is huge! I think this needs to be highlighted and explained.

We can add further discussion to this.

l. 274-275. Is this based on volumetric water content (Figs. 6c-d-e)? I think the surface runoff simulation would be interesting here. Consider adding them to Figure 7.

We will look at these results to see if they add anything to the presentation of results.

Figure 7. Units and date formats should be consistant with other figures (Figs. 4 to 6).

Will revise as suggested.

l. 283-284. In fact, ponding of liquid water at the base of the snowpack was not demonstrated by your results, but rather suggested by simulations and SWE observations. However, evidence of ponding could be provided by snow pit observations. If you have such observations of ponding at the base of the snowpack, consider adding them. Otherwise, please revise the wording of this sentence.

Will add text to specify.

l. 288. Just to be sure, by observational data, do you mean the SNOTEL station measurements?

Yes, we will add that to be clear.

l. 298-300. The comparison with the northern aspect remains speculative as there were no wind speed measurements taken there.

Good point. We will clarify that it is possible for increased wind, but uncertain.

l. 302-304: Have you applied any wind undercatch corrections to the forcing precipitation?

We will have to double check on this.

l. 309-310. This response may be enhanced by lateral flow over ice layers in the snowpack. See Eiriksson et al. (2013 - https://doi.org/10.1002/hyp.9666).

Only if the ice layers are thick enough to be continuous. The Eiriksson et al. (2013) paper actually found that ice lenses did not divert liquid water laterally very well.

l. 313-314 and Figure 8. This should be moved to the Results section.

Will revise as suggested.

l. 316. Do you have any temperature observations from your snow pit observations (even once) to support this?

We do have some limited temperature measurements that supports this that we can include.

l. 342. Can you elaborate on the prevalence of hydraulic barriers in the northern aspect snowpack rather than in the southern aspect snowpack?

Yes, we can elaborate further to include the north aspect slope, including evidence such as the multiple ice lenses observed in the snow pit.

Figure 9. This conceptual figure is interesting, but it is not based on field evidence. This should be clearly stated in the text.

Part (b) is based on field evidence and is not changed from Webb et al. (2018). We will clarify that this is how we are interpreting the multiple observations made for part (a) of the figure.

I recommend that the authors compare their results with those of Mazzotti et al. (2023 - https://doi.org/10.5194/hess-27-2099-2023)

This paper is certainly related and will be compared in the revisions.

l. 380-381. This has not been directly observed and remains a hypothesis. I would refrain from drawing conclusions from this.

Will revise to reflect this.

l. 384. Please add a few words on how these results would differ in different locations/climates. Please also add some concluding remarks on how the results of this work can improve our global understanding of snow in complex terrain and provide guidance for future research.

Will revise to include this, but our writing style is to have guidance for future research in discussion rather than conclusions.

Technical comments.

Will revise as suggested for all technical comments below.

l. 12, 20, 24 and so on… Please consider using the term "ponding" instead of "pooling" throughout your manuscript.

l. 12. This study measures --> In this study, we measured.

l. 15. input --> inputs

l. 16. models --> simulations

l. 16. missing word (that?)

l. 21. (snow) pit.

l. 31. 'Regional distributions in SWE also impact ecosystem services through surface albedo, effectively cooling earth surfaces and regulating climate'. It took me a few reads to understand this sentence. I

recommend the following change: 'Regional distributions in SWE also impact ecosystem services through surface albedo, which effectively cools Earth's surfaces and regulates climate.'

l. 41. measure --> estimate

l. 47. I do not get what you mean by "snow cover" being a snowpack properties.

l. 87. Please include the year of that reference

l. 97. Please include the year of that reference

l. 98. Please indicate that masl means meters above sea level

l. 100. Please verify the format of the date.

l. 127. were --> was

l. 153, un-necessary --> unnecessary (?)

l. 225-226. Please revise the syntax of this sentence

l. 265-266. Please, revise this sentence.

l. 266-267. Please indicating Fig. 7a-b only once.

l. 294. doesn't -->  does not

l. 318. Please remove "and requires further research in the future".

---

## Author Comment (AC2)

This study examines the influence of aspect and slope position on snowpack parameters i.e., depth, density, and liquid water content (LWC), within a subalpine watershed in Colorado, USA. The variations of these parameters are evaluated using GPR, in situ stations, snow pits and SNOWPACK modeling. The study found that mid-winter melt events predominantly affect south-facing slopes, triggering later flow of LWC downslope and the redistribution of SWE. Additionally, ice layers develop on south-facing slopes during mid-winter periods. Flat terrain exhibits a steady increase in soil moisture throughout the winter. In contrast, as spring progresses, north-facing slopes witness the pooling of liquid water at their base.

The findings underscore the importance of considering aspect and slope position when estimating snow water resources. However, many conclusions are based on qualitative reasoning and are not always support by the collected field evidence. While the snow modeling community is undoubtedly moving towards better representation of complex snow redistribution and melting processes, this paper does not provide sufficient quantitative evidence to significantly advance our current understanding of snow dynamics. If the authors intend to maintain a qualitative and conceptual approach, the manuscript should be retitled to reflect this focus. Additionally, a dedicated section should be included to address the study limitations. For instance, the paper could discuss why factors such as wind, canopy, terrain roughness, and eventually gravitational transport were not explicitly considered in this analysis.

Thank you for the comments. We appreciate the constructive suggestions and agree that further clarification on what is being interpreted versus directly observed would better represent this work. Additionally, more details on uncertainty and limitations can be expanded on during the revisions. Below are replies to specific comments in blue as well.

Major comments.

- While I appreciate the complexity of organizing extensive snow campaigns and the integration of various tools like GPR, snow pits, and SNOWPACK, I'm uncertain about the optimal utilization of GPR in this study. While GPR can efficiently survey transects, its application here seems to be limited to average this information to a single-point observations (derived from averaged TWT and snow depth along the transect). The potential uncertainty associated with this approach is not explicitly addressed, and it appears to be significant. Additionally, GPR limitations in wet snow conditions and its inability to provide detailed snow layering information, particularly regarding ice lens formation or wind redistribution, makes the use of GPR difficult to justify in this work. Furthermore, the absence of radargrams as supplementary materials, which is an interesting data per se, hinders reproducibility and future works.

These are good points that could use further description/justifications in the manuscript. We used GPR because of the smaller research team that would not be able to dig as many snow pits as GPR transects to cover the spatial extent of this research. However, we disagree that the GPR uncertainty is significant. Given that depth is well constrained through manual depth probes (more details on depth variability additions discussed in later comments/responses), we expect bulk SWE estimates to have similar uncertainty to pit observations (Meehan et al., 2024). We can certainly expand on this discussion for clarity in the revisions. We agree that it is a limitation in that GPR is unable to obtain detailed snow layering information. We are also happy to provide the radargrams as supplementary materials as well.

Reference: Meehan, T. G., Hojatimalekshah, A., Marshall, H.-P., Deeb, E. J., O'Neel, S., McGrath, D., Webb, R. W., Bonnell, R., Raleigh, M. S., Hiemstra, C., and Elder, K.: Spatially distributed snow depth, bulk density, and snow water equivalent from ground-based and airborne sensor integration at Grand Mesa, Colorado, USA, The Cryosphere, 18, 3253–3276, https://doi.org/10.5194/tc-18-3253-2024, 2024.

- The paper introduces the canopy influence as a key factor affecting the energy balance (L66 on), yet the specific role of canopy within the study domain remains unclear. While LiDAR data is mentioned and depicted in Figure 1e, its utilization in the analysis is not explicitly detailed. The discussion on canopy effects often lacks specificity, relying on generic considerations rather than relate to the specific test site. Similarly, the approach to estimating snow density from GPR data is confusing. The introduction suggests that density is generally considered uniform and that GPR can provide spatialized accurate measurements (L74 on). However, the subsequent averaging of density along transects contradicts this assumption. It would be beneficial to see a comparison of the radargrams, also at a qualitative level, before averaging them (this may further support the conceptual model of Fig 9). Additionally, the absence of uncertainty quantification in the results section hinders the interpretation of comparisons and the reliability of conclusions. I suggest addressing these points, such that the paper can strengthen its scientific rigor and provide a more comprehensive understanding of the complex interactions between canopy, topography, and snow processes.

Good point about the clarity of some of the methods. We will certainly revise some of the methods to clarify these points. LiDAR data were only used to characterize the site and canopy height in specific locations. Further details can be provided, but there was no use of LiDAR data in our analysis. We will also revise the text to clarify the GPR methods. We believe that density within each transect should be relatively uniform, but from transect to transect it will vary. The use of averaging is due to the different footprint of measurements between the depth probe and GPR. Uncertainty quantification can certainly be added to the manuscript to strengthen the rigor and provide further understanding.

Detail comments

L14 From Sec 2.3. it is not clear how the calibration of GPR snow density is done using snowpits and SNOTEL stations.

A more detailed description will be added. In general, we estimated the density using GPR-depth methods and compared to the snowpits and snow pillow, correcting for any bias with the assumption that the snowpits and pillow are the "true" values.

L23 This assertion seems to be limited to the particular characteristics of the study area and may not generalize to other conditions.

This is true, we will revise the text to clarify that this may be site-specific and further detail the site characteristics so other researchers may draw insights towards other sites.

L75 Typically, bulk snow density is measured using a federal tube or within snow pits by summing the density derived by smaller volume tubes (or triangular prisms), as described by Kinar and Pomeroy, 2015.

Yes, we can revise the text for clarity.

L91 Snow depth can vary significantly, even over short distances, due to the rugged and heterogeneous nature of alpine terrain. This variability, combined with the small area sampled by a probe, highlights the importance of quantifying uncertainties in snow density estimates. Generally an average of N measurements should be done.

Yes, we averaged a minimum of 8, but generally at least 10 probed depths, at 2 meter spacing. We will add the Lopez-Moreno et al. (2011) citation as well as conduct uncertainty analysis based on this. We will also provide more details on the measured depths in supplementary material that includes standard deviations of each transect. To summarize the observed variability of the 32 transects measured, the standard deviation of depth observations ranged from 4 cm to 25 cm with an average of 10.6 cm and a median of 9 cm.

Reference: López-Moreno, J. I., Fassnacht, S. R., Beguería, S., and Latron, J. B. P.: Variability of snow depth at the plot scale: implications for mean depth estimation and sampling strategies, The Cryosphere, 5, 617–629, https://doi.org/10.5194/tc-5-617-2011, 2011.

L92 If the primary focus of the research is to investigate the impact of aspect and slope position on snowpack dynamics, a thorough justification is required to explain why factors such as wind, canopy, terrain roughness, and gravitational transport were not explicitly considered in the study, especially given their potential influence on snow distribution and melt.

A thorough justification for this will be given in the revisions, as well as discussion towards these limitations for the study.

Fig 1a please rotate it consistently with the other figure (i.e., North up)

The other reviewer mentioned this as well. We will revise in this manner.

L162 Please explicitly state that, as reported in Webb & Mooney 2024c, TWT is calculated as an average value.

Will revise to be more explicit.

L170 the equations must be numbered.

Thank you for pointing this out. We will revise this.

L175 Please provide a method for calculating the uncertainty associated with the TWT measurements. Given the potential for significant error propagation due to small denominator values, a rigorous uncertainty analysis is essential.

An uncertainty analysis will be conducted and presented in revisions. We agree that it has the potential to be quite significant, especially under shallow snow conditions.

Section 2.4 how the SNOWPACK free parameter has been calibrated?

I am not sure what is meant by the free parameter, but in line with another reviewer's comments more details on the modeling methods will be given. The input files will also be made available in the supplementary material for reproduction of the work. There was minimal calibration of SNOWPACK, though, as the focus was to determine if and when surface melt events were occurring to support some of the interpretation of observations we made in the field.

Figure 5 is difficult to interpret. A simpler, more traditional visualization would improve the comparison of differences between the data.

Will revise to a more traditional plot organization.

Figure 6 please report the uncertainty for all the measurements.

Will report in revisions.

L287 "unusual results" respect what?

Will revise for clarity.

L305 "model weakness"? Can you better elaborate the sentence?

Will elaborate in revisions.

L308 Can you better justify this sentence showing the evidence of this mechanism?

Yes, these mechanisms will be further discussed and linked to the evidence that was observed. We will also be more clear that some of these are interpretations and not directly observed.

L 333 Why "unrealistic"? Can you better elaborate it?

Yes, the derived density was greater than 1000 kg/m$^3$ (the density of water). We will elaborate further in the revisions for clarity as well as explain why this happens in GPR data.

L 354 The answer to the main research question of the paper is answer considering only the melting. So, the melting was the focus of the research?

The focus did become melting. We can further clarify and revise the question and title to reflect this.

Figure 9. This conceptual figure is interesting, but it is not based on field evidence. This should be clearly stated in the text.

We can add further details and link the interpretation to observations. But, you are correct that this is an interpretation that may be better presented and stated as a hypothesis with alternative hypotheses as possibilities. We will revise in this manner.

L 367 I suspect that Dingman simplified his modeling to a homogeneous snowpack. While the four-phase model remains valid for individual homogeneous layers, additional complexity is necessary to accurately represent real-world snowpacks (which however is made up of different homogeneous layer, possibly at different phase).

This refers to the 4 phases: accumulation, warming, ripening, output. Revisions will clarify this.

L371 Given the significant spatial variability in snow depth, particularly in complex terrain, it is challenging to believe that traditional probing methods can accurately capture these variations without averaging N measurements and without a rigorous uncertainty analysis.

As mentioned above, an uncertainty analysis will be conducted. However, the survey was designed based on the Lopez-Moreno reference also given above. More data would have likely been better (as it always is), but the actual variability of the depth could not be known until after the data were collected.

L374 and conclusion: So this is only a study on the energy balance and not on snow redistribution processes?

The redistribution refers to the interpretation of SWE moving down the hillslope through lateral flow paths. This will be clarified in revisions and other terminology considered to avoid confusion.

As a final note, while there are no explicit publisher guidelines against self-citation, it is generally advisable to minimize excessive self-referencing. For instance, the accurate prediction of LWC by SNOWPACK could be supported by citing previous studies (as done in the current self-cited works) that provide also detailed information about the model details, which is not developed by the authors.

Thank you for pointing this out. It is easiest to cite one's own studies because sometimes they come to mind first. But, this is good advise that I agree with. Some of these will be replaced during revisions to avoid over self-citing.

The References section is difficult to read due to the lack of spacing between entries. Additionally, some references appear to be formatted incorrectly e.g., L87 Clark et al. should be Clark et al., 2015.

Formatting will be double-checked during revisions throughout the text, but the journal manuscript template was used with respect to the references section spacing. We can confirm the formatting of this section and revise if mistaken.

Kinar, N. J. and Pomeroy, J. W.: Measurement of the physical properties of the snowpack, Rev. Geophys., 53, 481–544, https://doi.org/10.1002/2015RG000481, 2015.

**Citation**: https://doi.org/10.5194/egusphere-2024-2364-RC2

---

## Author Response (AR2)

Editor Comments:

Overall, the manuscript was largely improved and I am convinced of the importance of this study.

In a next round, please consider the comments of both reviewers, and in particular the comments of reviewer 2 regarding (discussing) potential constraints and strengthening the statistics and uncertainty analysis.

I am looking forward to the revised manuscript addressing the comments in detail.

Best regards,

Franziska Koch (Associate Editor, The Cryosphere)

Reviewer 1: Benjamin Bouchard

The authors have substantially improved their manuscript from the first version submitted to The Cryosphere. They have addressed the vast majority of comments raised by the reviewers. In this sense, I recommend this version of the manuscript for publication, subject to one clarification. In Appendix C, the authors report a multiplication factor applied to RAWS data for shortwave radiation and refer to the main text for explanations (see captions of Tables C2, C3 and C4). However, the justification for the multiplication is missing from the manuscript. Before publication, I would like this point to be addressed and the multiplication factor clearly explained in the manuscript.

Thank you for pointing this out. We have added the following text to lines 225-226:

"Incoming shortwave radiation for each simulation used the RAWS station data and a location specific multiplication factor determined by the 1 Mar solar radiation model from the 1-meter DEM (Fig. 1c)."

Reviewer 2

I appreciate the authors revisions and find the topic of this study highly relevant. However, I maintain my concern regarding the reliance on qualitative reasoning in drawing conclusions, which often lacks robust support from the field data.

Specifically, the following points require further attention:

The study scope (and title), encompassing five campaigns over a single melting season (five months), presents a limited dataset. This constraint significantly affects the robustness of your conclusions, particularly regarding the complex interactions being investigated in a partially forested area (the observations are of SWE and density but the main driving process seems to be the isolated by the authors in the lateral flow of liquid water). Indeed, the influence of the forest canopy on snow processes is a critical factor, especially considering the presence of at least three

transects within forested areas. The manuscript would benefit from a more rigorous approach to disentangling the effects of canopy cover from other variables. Currently, it is unclear how the observed increases in snow density and SWE at the North and South base sites (inside the forest) during April and May were isolated from canopy influences.

We agree that it is not entirely isolated from canopy influences, though there will generally be denser canopy where there is more water available for vegetation to grow so it is difficult to fully disentangle these processes. We have modified the text (lines 415 – 424) to better reflect this:

"Canopy and wind drifting influences are not interpreted as major contributions to the redistribution of SWE at Dry Lake due to the wind direction being parallel to the study slope contours and observations of drifting not occurring at our transect locations. However, some canopy shading will influence the observations at the base of the north aspect hillslope though terrain shading dominates the energy balance as previously mentioned. Further, the deciduous canopy at the base of the south aspect has been observed to have more SWE than on the hillslope (Webb, 2017), but under canopy conditions here generally have lower snow density whereas we observed an increase in density at the base of the south aspect slope. Thus, we interpret the lateral flow of liquid water to be an important factor in the redistribution of SWE as presented in the perceptual model for the Dry Lake site (Fig. 8), though we are unable to fully disentangle the influence of canopy and lateral flow from one another."

The derivation of uncertainty requires a more rigorous formulation. As it stands, the lack of a clear statistical framework hinders the comparison of density measurements. I strongly recommend a more robust approach, such as: calculating the standard deviation of the snow depth measurements along the transect. Propagating this uncertainty to estimate the uncertainty in the averaged density and SWE. This will provide a more transparent and statistically sound basis for comparing measurements and assessing the significance of observed differences.

Currently, the manuscript implies differences in density and SWE between locations. However, without a proper uncertainty analysis (also in the plot expressed as uncertainty bars), it is difficult to determine if these differences are statistically significant.

To enhance the scientific rigor of the paper and provide a more comprehensive understanding of the interactions between canopy, topography, and snow processes, I suggest the authors address these points.

It is well-understood that forest canopy will be denser where plant available water is more abundant. This will happen, for example, on north facing slopes where more snow accumulates. We mention this in lines 356 – 358: "This difference in canopy cover is likely attributed to aspect as the deeper snow and increased soil moisture on the northern aspect increases the amount of plant available water for vegetation growth". Thus, it is not possible to fully disentangle the role of canopy from the processes. We have modified the text as quoted in the previous comment to better reflect this.

For the uncertainty, we have revised the analysis as suggested to use standard deviations with revisions as follows.

Lines 179 – 186:

**2.4 Uncertainty Estimation of Survey Data**

The above-described methods in estimating $\rho_s$ and SWE require additional estimates of uncertainty. We used the standard deviation ($\sigma$) of the GPR TWT and manually measured $d_s$ data to estimate the uncertainty associated with the derived values using equations (1), (2), and (3). Due to $d_s$ and GPR TWT being correlated to one another and not independent, we estimate the range of $v$ through:

$$v_{+\sigma} = \frac{d_s + \sigma_{ds}}{\dfrac{TWT + \sigma_{TWT}}{2}} \tag{4}$$

$$v_{-\sigma} = \frac{d_s - \sigma_{ds}}{\dfrac{TWT - \sigma_{TWT}}{2}} \tag{5}$$

where $v_{+\sigma}$ and $v_{-\sigma}$ are the $v$ calculated with variables plus or minus their associated $\sigma$, respectively; and $\sigma_{ds}$ and $\sigma_{TWT}$ are the $\sigma$ associated with $d_s$ and GPR TWT, respectively. These values of $v_{+\sigma}$ and $v_{-\sigma}$ were then used to propagate this variability through equations (2) and (3).“

New Figure 4:

[Figure]

Figure 4: Results from the transect observations including calculated SWE, observed $d_s$, and GPR derived $\rho_s$ for: a) Flat Terrain around the SNOTEL station, b) North Top, c) North Middle, d) North Base, e) South Top, f) South Middle, and g) South Base locations. Pit measured average densities are shown when collected, and SNOTEL station data are displayed for additional

comparisons of SWE and $\rho_s$. Uncertainty bars are shown for SWE and GPR derived $\rho_s$ using $\sigma$ of collected data. Note that the GPR results that gave unrealistic values due to the presence of liquid water is slightly greyed in panel d.